# Early high rates and disparity in the evolution of ichthyosaurs

Benjamin C. Moon [1✉] & Thomas L. Stubbs [1]

How clades diversify early in their history is integral to understanding the origins of biodiversity and ecosystem recovery following mass extinctions. Moreover, diversification can represent evolutionary opportunities and pressures following ecosystem changes. Ichthyosaurs, Mesozoic marine reptiles, appeared after the end-Permian mass extinction and provide opportunities to assess clade diversification in a changed world. Using recent cladistic data, skull length data, and the most complete phylogenetic trees to date for the group, we present a combined disparity, morphospace, and evolutionary rates analysis that reveals the tempo and mode of ichthyosaur morphological evolution through 160 million years. Ichthyosaur evolution shows an archetypal early burst trend, driven by ecological opportunity in Triassic seas, and an evolutionary bottleneck leading to a long-term reduction in evolutionary rates and disparity. This is represented consistently across all analytical methods by a Triassic peak in ichthyosaur disparity and evolutionary rates, and morphospace separation between Triassic and post-Triassic taxa.

[1] Palaeobiology Research Group, University of Bristol, Life Sciences Building, 24 Tyndall Avenue, Bristol BS8 1TQ, UK ✉email: benjamin.moon@bristol.ac.uk

Understanding the expansion of biodiversity in living and fossil groups is a fundamental objective of evolutionary biology. The early burst (EB) model has received considerable attention in recent years[1–3]. As originally envisaged by Simpson[4,5], the model predicts rapid and extensive morphological diversification early in the evolutionary history of clades. This results from high evolutionary rates and gives rise to early maximal morphological diversity (disparity), followed by a marked slow-down in rates and reduced disparity. The prevalence of EB has been mixed in numerical studies[6]. Phylogenetic comparative methods often fail to identify EB patterns in living groups[1], while in fossil disparity studies EB trends are more common[2], although not universal[7], and many studies consider disparity patterns without incorporating quantitative tests of the underlying rate processes[2,8].

The ecological and geological contexts of evolutionary diversifications have emerged as key parameters for testing EB trends. Diversifications post-mass extinction should, in theory, provide the ideal conditions[9]. When the evolutionary landscape is perturbed and potential competitors are removed from ecosystems, new ecological opportunities may arise for expanding clades[4,10,11]. Major ecological transitions, such as from land to sea[12], the evolution of flight[13], or new feeding innovations[14], can also open up previously unexplored niche space—the specific size, habitat, feeding, locomotory, and other traits of an organism—and may serve as catalysts for EB diversifications.

Ichthyosaurs (Ichthyosauriformes), an iconic group of extinct fish-shaped marine reptiles, represent an excellent case study for testing the EB model. The group diversified in the aftermath of the Permo-Triassic mass extinction (PTME) and, along with sauropterygians, represent the first major radiation of tetrapods in the marine realm that show specialised adaptations to aquatic environments[15]. The faunal recovery interval following the PTME was an exceptional time in the diversification of life, and ichthyosaurs expanded into trophic niches that had not been occupied by tetrapods in the Palaeozoic[16]. Ichthyosaurs became important components of marine ecosystems and persisted for around 160 million-years (my), until their extinction in the mid-Cretaceous. Previous morphological disparity studies have highlighted an evolutionary bottleneck in ichthyosaur evolution: during the Late Triassic the clade was reduced to a small number of lineages and this led to a long-term reduction in disparity, but species diversity recovered[17].

No previous studies have explored evolutionary rates in ichthyosaurs throughout their entire evolutionary history and specifically tested for EB using state-of-the-art quantitative methods. In the past, EB-modelling approaches have generally relied on whole-tree transformations that assume a single process across the whole group[1] and may underestimate rate variation[6,18]. Here we implement more flexible modelling approaches that test for rate heterogeneity based on individual branch modifications[19]. We use a combined disparity and rates approach to examine both the pattern and process of morphological evolution in ichthyosaurs. We analyse a large morphological dataset that characterises skeletal variation as discretely coded characters[20], and consider skull length as a complementary but independent metric of variation that serves as an approximation for overall size. We calculate disparity and evolutionary rates through time to test for early maximal morphological diversity and early high evolutionary rates. Temporal morphospace trends are explored to test if disparity and rate patterns result from an exhaustion or saturation of morphologies[3,21], or if ichthyosaurs continued to innovate throughout their evolution.

Our analyses show that ichthyosaur morphological disparity and evolutionary rates peaked early in their history during the Triassic—an archetypal EB pattern. Following substantial loss of morphological variation, and the survival of a single lineage after the Triassic, ichthyosaurs have a much restricted morphospace occupation and low rates of evolution during the last 100 my of their history, and a protracted stagnation in their evolution.

## Results

**Morphological disparity.** We used a discrete skeletal character data set of 114 Ichthyosauriformes for this analysis[20]. The character data were converted to a taxon–taxon distance matrix using the package Claddis[22] in R version 3.6.1[23] (see the section "Methods" below). Based on maximum observed rescaled distances (MORD)[24,25] we used two main measures of disparity: pairwise distances and weighted pairwise distances[26] calculated per-bin using two binning schemes: epoch-length bins and equal 10 my bins[17]. Additionally, we used principal coordinates analyses to reordinate the distance matrix and produce morphospace plots for Ichthyosauriformes.

Morphological disparity rapidly accumulated early in ichthyosaur evolutionary history. Disparity peaked in the Middle and Late Triassic, followed by a long-term decline through the Jurassic and Cretaceous (Fig. 1; Supplementary Figs. 1–5). Both weighted and un-weighted pairwise dissimilarity produced similar trends. The earliest Triassic bin has similar or higher disparity than the post-Triassic bins despite the shorter length. Using equal-length bins throughout distinctly reduces the resultant disparity of post-Triassic bins relative to those in the Triassic (Fig. 1b). Disparity is highest in the first four 10-my-bins (251.3–211.3 Ma), particularly in those representing the early Middle Triassic and middle Late Triassic. No certain ichthyosaur taxa are present between 211.3 and 201.3 Ma, so disparity for this bin cannot be calculated (Fig. 1b). There is a further disparity decline in the Early Cretaceous that is only visible using equal-length bins.

Triassic ichthyosaurs occupied a larger part of the morphospace than post-Triassic taxa despite the difference in timespan (Fig. 2; Supplementary Fig. 6). Morphospace occupation broadly overlaps between Early–Middle Triassic taxa and Middle Jurassic–Late Cretaceous taxa, respectively. A distinct separation found between Late Triassic–Early Jurassic taxa supports the evolutionary bottleneck described by Thorne et al.[17], but is bridged somewhat by the taxa from each of these bins (Fig. 2; Supplementary Fig. 6). Negative eigenvalue correction during the multivariate ordination has little effect on the morphospace occupation of the first three principal coordinates axes, but reduces the variance described considerably.

Pairwise PERMANOVA tests for morphospace separation found significant differences between the Triassic bins and all others in almost all cases ($p < 0.05$; Supplementary Code 1). Non-significant differences were always found between Middle Jurassic–Late Cretaceous bins, and between the Middle–Late Triassic and Late Cretaceous bins. Statistical tests also confirm that disparity varied significantly throughout ichthyosaur evolutionary history. Significant differences in disparity were found consistently between the Early–Middle Triassic, Late Triassic–Early Jurassic, and Early–Late Cretaceous epoch-bins (pairwise $t$-tests, $p < 0.05$; Supplementary Code 1). For the consecutive 10-my-bins, significant differences in disparity were consistently found between bins 1–2 (~Early–Middle Triassic), 8–9 (~Toarcian–Middle Jurassic), 10–11 (~Oxfordian + Kimmeridgian–Tithonian + Berriasian), and 12–13 (~Valanginian + Hauterivian–Barremian + Aptian). Disparity in bins 2–4 (~Middle–Late Triassic) is mostly significantly higher than in bins 9–16 (~Middle Jurassic–Late Cretaceous), and disparity in bins 6–8 (~Early Jurassic) is significantly higher than in bins 9–12 and 14–16 (~Middle Jurassic–Cretaceous, excluding bin 13 [~Barremian–Aptian]).

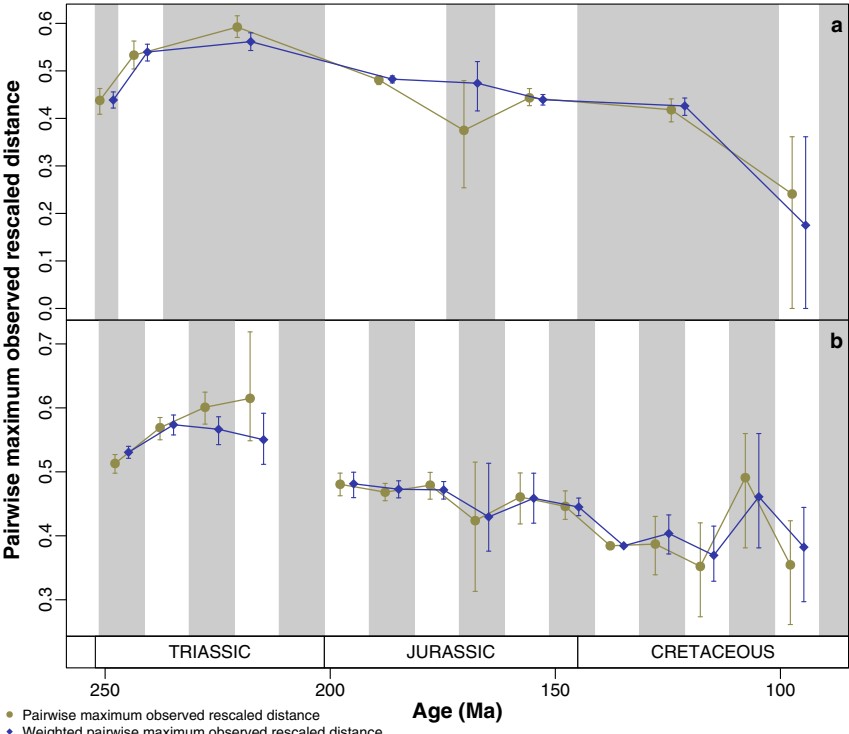

**Fig. 1 Per-bin discrete skeletal character disparity of Ichthyosauriformes through the Mesozoic. a** Per-epoch disparity of ichthyosaurforms. **b** Per-equal-length 10-my-bin disparity of ichthyosauriformes. Pairwise and weighted pairwise dissimilarity measured from the maximum observed rescaled distances between ichthyosauriform taxa in the dataset of Moon[20]. Mean values and 95% confidence intervals are shown from 500 bootstrap replicates for each bin.

**Discrete character evolutionary rates**. To examine rates of evolution we time-scaled 100 phylogenetic trees sampled from the Bayesian posterior distribution of Moon[20] using the Hedman scaling method[27,28]. Rates of discrete character evolution were calculated for each tree using Claddis[22] in R version 3.6.1[23] as per-bin values in epoch-length and 10-my bins (see the "Methods" section below).

Rates of discrete character evolution in ichthyosaurs are heterogeneous (Fig. 3; Supplementary Fig. 7) and have a consistent pattern of high early rates followed by longer-term slow rates of evolution. Rates were fastest in the first 10 my of ichthyosaur evolution (Fig. 3b). Averages across the 100 analysed phylogenetic trees decline precipitously through the earliest bins (~40 my) in all analyses, but then remain stable from the latest Triassic through to the extinction of the ichthyosaurs 120 my later. All trees show significantly higher-than-background rates in the Early–Middle Triassic (first two epoch-bins and 10-my-bins; Fig. 3). Few trees show rates that increase between the Early–Middle Triassic and none do between the first two 10-my-bins. However, discrete rates of evolution decrease into the Late Triassic so that no trees show significantly high rates by the Jurassic (Fig. 3), a turnaround within 30 my. Significantly slower-than-background rates occur earliest after 221.3 Ma (Fig. 3b), or in the Early Jurassic epoch-bin (Fig. 3a), and throughout many—but not all—trees and bins in the Jurassic and Cretaceous.

**Skull size evolutionary rates**. To examine trait evolution more directly we collected skull length data for 64 ichthyosauriform species and the outgroup *Hupehsuchus nanchangensis*. Rates of skull size evolution were analysed using the variable rates model in BayesTraits version 2.0.2[19,29] and R[23] using 100 Hedman time-scaled phylogenetic trees (see the "Methods" section below). We examined rate heterogeneity across ichthyosauriform phylogeny and through time for each dated tree using equal 10-my time slices.

Ichthyosauriformes rapidly diverged into a broad range of skull sizes in the first 40 my of their evolutionary history, followed by a stepwise reduction in size disparity through the Jurassic and Cretaceous (Fig. 4a). Skull size evolutionary rates were highly heterogeneous. Across the 100 analysed phylogenies, the variable (heterogeneous) rates model received overwhelming support; 99% showed very strong evidence for heterogeneity (logBFs > 10) and 100% had strong evidence (logBFs > 5). Results are consistent across all 100 tested phylogenies (Fig. 4b; Supplementary Fig. 8). High rate branches are concentrated in Triassic ichthyosaurs, including basal taxa (e.g. *Chaohusaurus, Utatsusaurus, Xinminosaurus, Thalattoarchon*), Mixosauridae, and Shastasauria. High rates are seen in some Early Jurassic taxa, such as *Temnodontosaurus* and Leptonectidae. But overall, low evolutionary rates characterise a large proportion of Jurassic and Cretaceous Neoichthyosauria, including all members of Ophthalmosauridae (Fig. 4d). When phylogenetic branches are scaled according to evolutionary rate, rather than geological time, it highlights the clear stagnation of phenotypic evolution (Fig. 4d). Rates through time confirm a distinct EB trend (Fig. 4c). The first 40 my of ichthyosauriform evolutionary history show notably fast rates of skull size evolution, particularly in 10-my-bins 2 and 3, corresponding to the late Middle and early Late Triassic. There is a sharp rate reduction during the latest Triassic. In the Early Jurassic rates were moderately low, and for the last 70 my of ichthyosauriform evolutionary rates became exceptionally low.

## Discussion

Ichthyosaur evolution describes a classic example of an EB model. The combination of early high disparity and rapid evolutionary rates for both discrete characters and skull length indicates that ichthyosaurs burst onto the scene, rapidly diversified phenotypically, and adapted to numerous ecological niches[30] (Figs. 1–4; Supplementary Figs. 1–8). While these results support previous

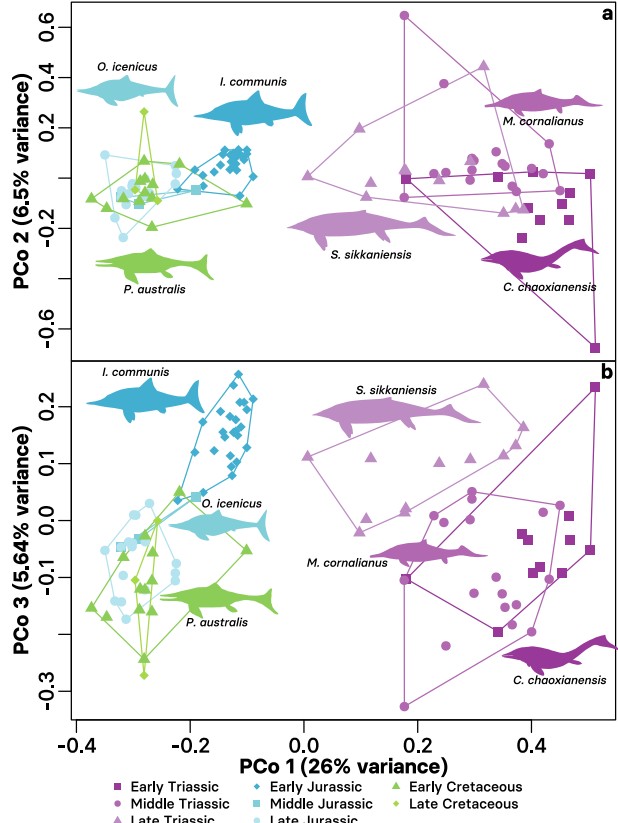

**Fig. 2 Morphospace occupation of Ichthyosauriformes through the Mesozoic. a** Principal coordinate axis 1 (26.0% of total variance) against axis 2 (6.5% of total variance). **b** Principal coordinate axis 1 against axis 3 (5.64% of total variance). Ordinated maximum observed rescaled distance matrix calculated from the cladistic data set of Moon[20] binned into epochs. Silhouettes of indicative taxa (*Grippia longirostris* [D. Bogdanov, vectorized by M. Keeley], *Ichthyosaurus communis* [G. Monger], *Mixosaurus cornalianus* [G. Monger], *Ophthalmosaurus icenicus* [G. Monger], *Platypterygius sachicarum* [Zimices], and *Shastasaurus sikkaniensis* [G. Monger]) show their relative positions. Images downloaded from PhyloPic and used under a CC-BY 3.0 licence: https://creativecommons.org/licenses/by/3.0/.

studies on the evolution of morphological diversity in ichthyosaurs[17], the identification of early high rates is novel.

Our analyses may even underestimate the magnitude of early high evolutionary rates seen in ichthyosaurs. The Hedman phylogenetic time-scaling protocol we used did not enforce a maximum age for the clade[27]. In our sample of dated trees, the root age ranged from 256.1 to 254.8 Ma, pushing the clade's origins into the Late Permian (Lopingian). Ichthyosaur origins remain enigmatic and age of the clade has been debated, but there are no fossil occurrences prior to the Early Triassic[31]. If we enforced a strict a priori maximum age constraint of the earliest Triassic this would reduce the branch lengths for the earliest diverging forms. Branch lengths are an essential parameter in rate calculations[19] and shorter branch lengths would inevitably generate higher rates.

Despite these high initial rates of evolution, there appears to be stepwise appearances of ichthyosaur baupläne through the Triassic (Figs. 2, 4; Supplementary Figs. 6, 7). The earliest ichthyosaurs (e.g. *Chaohusaurus, Utatsusaurus*) represent the basal, most lizard-like condition: elongate body, partially developed paddle-like limbs, poorly differentiated caudal fluke (if any)[32]; the basal grade of ichthyosaur evolution[20,33]. These taxa were rapidly replaced by intermediate grade ichthyosaurs in the Middle

Triassic (e.g. *Cymbospondylus, Mixosaurus*)[20,33]; however, the more basal of these taxa occupy a similar morphospace region to earlier taxa (Fig. 2). Recent discoveries from the Early–Middle Triassic, particularly from China, have added greatly to the known diversity and ecologies of early ichthyosaurs[34,35]. In particular, the early radiation of ichthyosaurs included putative amphibious and nearshore taxa (e.g. *Cartorhynchus, Mixosaurus*) alongside larger predatory ichthyosaurs (e.g. *Cymbospondylus, Thalattoarchon*)[34,36], which suggests the formation of a complex, multilevel ecosystem within ~8 my of the Permian–Triassic boundary, and only 4.5 my after the earliest occurrence of Ichthyosauriformes[31,34,37]. These ecologies are not represented later in the evolution of ichthyosaurs, which are limited to only open ocean forms.

The morphological data analysed here does not code specifically for ecology (in fact characters actively avoid doing so ref. [20]), yet many characters do encapsulate the manifest morphological changes that occur in the ecological transitions that these ichthyosaurs undergo. This suggests that the evolution of these traits was somewhat gradual as ichthyosaurs explored the morpho- and ecospace emptied after the PTME[16]. This gradual modification of morphospace contrasts previous results that found a rapid early shift in ichthyosaur ecospace occupation[30]. We suggest that this may result from two main causes: the characters used by Dick and Maxwell[30] draw particular focus to the changes that occur in Triassic ichthyosaur taxa—in particular body size dominates, and feeding strategy (ambush/pursuit) creates substantial sub-setting between Early–Middle Triassic and Late Triassic–Cretaceous taxa; and the inclusion in the cladistic data[20] of more specific osteological characters further separates species that may have otherwise similar ecologies.

A key period of transition occurs in ichthyosaur evolution between the Late Triassic–Early Jurassic. Then the diversity of intermediate-grade Triassic ichthyosaurs were replaced by thunniform-grade Neoichthyosauria[20], which are the only clade to survive across the Triassic–Jurassic boundary following an evolutionary bottleneck[17]. Our results represent this transition by a substantial separation in morphospace between the Late Triassic and Early Jurassic bins (Fig. 2; Supplementary Fig. 6), and significant decrease in disparity (Figs. 1, 4a; Supplementary Figs. 1, 4). Unfortunately, the latest Triassic record of ichthyosaurs is poor; the high Rhaetian skeletal completeness identified by Cleary et al.[38] is attributable to representation by *Leptonectes tenuirostris*, which is known from highly complete specimens in the Early Jurassic[39]. However, Rhaetian ichthyosaurs are present and demonstrate a high diversity of taxa and ecology, but incompleteness and uncertain taxonomic affinities has limited their inclusion in analyses[40,41]. This suggests extensive turnover throughout the later part of the Triassic, but a more rapid extinction at the Triassic–Jurassic boundary; the selectivity of this extinction in ichthyosaurs has yet to be tested. Recent finds indicate that this is consistent in other marine tetrapod clades that also cross the boundary (e.g. Sauropterygia[42]).

The success of ichthyosaurs following the Triassic was mixed: while the taxic diversity of ichthyosaurs peaked in the Early Jurassic, the disparity of the clade and rates of evolution decreased into the Jurassic and throughout the 100 my that ichthyosaurs continued to exist (Figs. 1, 3, 4; Supplementary Figs. 1, 4, 8). Despite increased completeness and comparability of specimens available due to extensive lagerstätte deposits—particularly in north-western Europe (e.g. Lias Group, Posidonia Shale Formation)—the disparity of Early Jurassic ichthyosaurs is less than the substantially less complete Middle–Late Triassic taxa[24,38]. The evolution and diversification of Sauropterygia and Thalattosuchia in the Early Jurassic likely increased competition between these taxa and ichthyosaurs[16], yet at this time

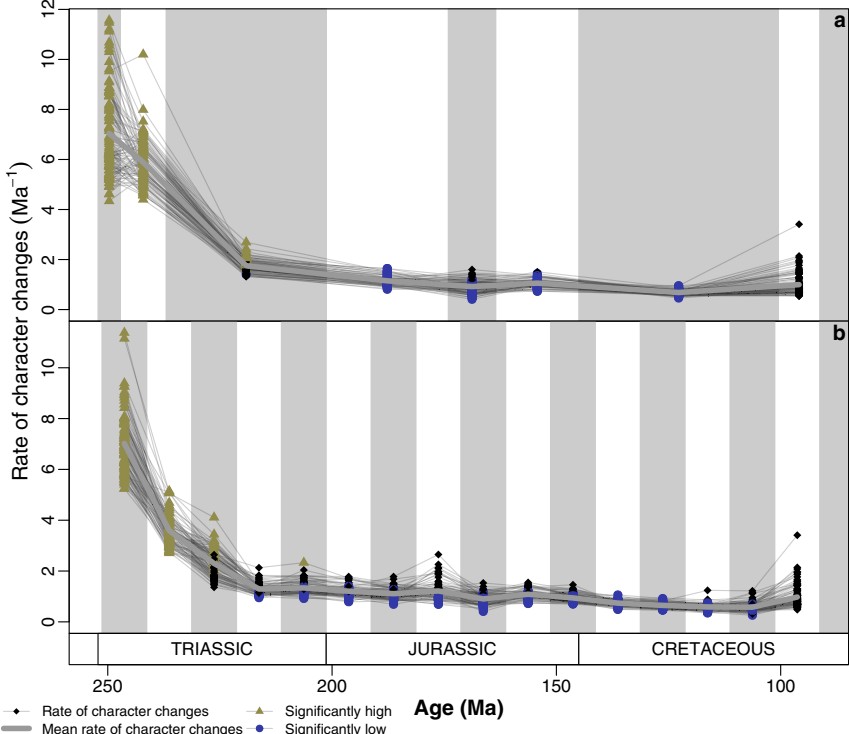

**Fig. 3 Rates of discrete skeletal character evolution in Ichthyosauriformes. a** Per-epoch rates of discrete character evolution. **b** Per-equal-length 10-my-bin rates of discrete character evolution. Calculated from the matrix of Moon[20] using 100 trees time-scaled using the method of Hedman[27]. Each line illustrates the results from a single tree. Significantly high (green triangles) and low (blue circles) rates of phenotypic evolution are highlighted. Black diamonds represent rates that are not significantly different from the pooled average. Thickened grey line shows the mean per-bin rate across all analyses.

ichthyosaurs exhibit a notable disparity (Figs. 1, 2, 4a; Supplementary Figs. 1, 4, 7) and diversity of ecology[30], including the swordfish-like *Eurhinosaurus* and the huge hypercarnivore *Temnodontosaurus*. Following this early dominance, ichthyosaurs become a relatively less important component of marine reptile ecosystems through the Jurassic[16], accompanying their decline in diversity and disparity (Figs. 1, 4a), and conserved morphospace occupation throughout much of the Jurassic and Cretaceous (Fig. 2). High-resolution study of a single evolving ecosystem demonstrates how later Jurassic ichthyosaurs became marginalised ecologically based on feeding guild[43].

Alongside this lack of diversity and disparity in post-Triassic ichthyosaurs is a substantial decrease in the rates of evolution in skull length and discrete character changes (Figs. 3, 4; Supplementary Fig. 8). Previously, a decline has been identified in ichthyosaurs through the Cretaceous[44], however, our results extend this back substantially to the Early Jurassic. We attribute this to increased taxon sampling and argue that this gives a more complete picture of ichthyosaur evolution at this crucial time. Nonetheless, there is a heightened decrease in disparity across the Jurassic–Cretaceous boundary, associated with substantial turnover[20] (Figs. 1 and 2); which is consistent with other marine reptile clades[45,46]. Fischer et al.[44] highlighted the potential link between environmental volatility, slow evolutionary rates, and ichthyosaur extinction in the Cretaceous. The long-term stagnation of ichthyosaur phenotypic evolution identified here complements this hypothesis.

## Methods

**Taxon sampling**. All 114 valid Ichthyosauriformes and one outgroup taxon were included from the phylogenetic analysis of Moon[20]. Characters coded for each taxon were based upon all reasonably assigned material, including published and confidently referred species, not just on holotypic specimens (data sources are discussed in Moon[20], supplemental document S.2). Completeness varied greatly

between 1.39% for *Cymbospondylus piscosus* to over 95% in *Ichthyosaurus communis* and *Ophthalmosaurus icenicus*. Occurrences of Ichthyosauriformes were taken from the primary literature. Where possible, first and last appearance dates (FADs and LADs) of taxa were recorded to ammonite or conodont biozone level. Occurrences were converted to absolute dates using Gradstein et al.[47] (Supplementary Table 3). Two time-binning schemes were used: eight epoch-level bins that correspond to the bins used previously by Thorne et al.[17], however these are of uneven lengths; and 16 bins of 10 my each, with the bins aligned so that the Triassic–Jurassic boundary falls at a boundary between bins (201.3 Ma, earliest bin starts at 251.3 Ma; Supplementary Table 1). The earliest Ichthyosauriformes are known definitely from the later Olenekian (248.8 Ma)[31], which the start of the earliest bin precedes in both schemes. Aligning the bins to different events allows more explicit analysis of the effect of these on the evolution of Ichthyosauriformes, while maintaining equal-length bins for comparison. The base of the Jurassic was chosen as this is the point identified by Thorne et al.[17] as a major turnover or bottleneck in ichthyosaur evolution.

**Discrete character data and disparity analyses**. Calculations of cladistic disparity used the packages Claddis[22] and dispRity[48] in R version 3.6.1[23]. The discrete character matrix of Moon[20] was converted to a distance matrix using the maximum observable rescaled distances (MORD) conversion of Lloyd[22]. Ancillary calculations used Gower (GOW), raw Euclidean (RAW), and generalised Euclidean (GE) distance conversions (Supplementary Fig. 1), but the latter two are susceptible to fossil record incompleteness and can lead to morphospace centroid slippage when using poorly coded taxa[24,25,49]. Disparity analyses calculated per-bin pairwise distances and weighted pairwise distances based on taxon completeness directly on the distance matrices using functions modified from Close et al.[26]; mean values and 95% confidence intervals were calculated from 500 bootstrap replicates. Weighted pairwise distances were calculated using the formula $\frac{\sum (D \times C)}{\sum C}$, where $D$ is the upper triangle of the matrix of pairwise distances and $C$ is the upper triangle of the matrix of comparable characters[26]. We further rarefied the data in each time bin to explore the effect of sample size (Supplementary Fig. 2).

To visualise morphospace occupation, the morphological distances were ordinated using principal coordinate analysis (PCOa) both with and without correction for negative eigenvalues[50] (to compare eigenvectors and eigenvalues). Fourteen taxa with incomparable characters were removed from MORD, GOW, and RAW matrices using the function TrimMorphDistMatrix of Claddis, and additionally from GE matrices for parity; most taxa removed have <20% characters coded (completeness). Additionally, the outgroup taxon, *H. nanchangensis*, was

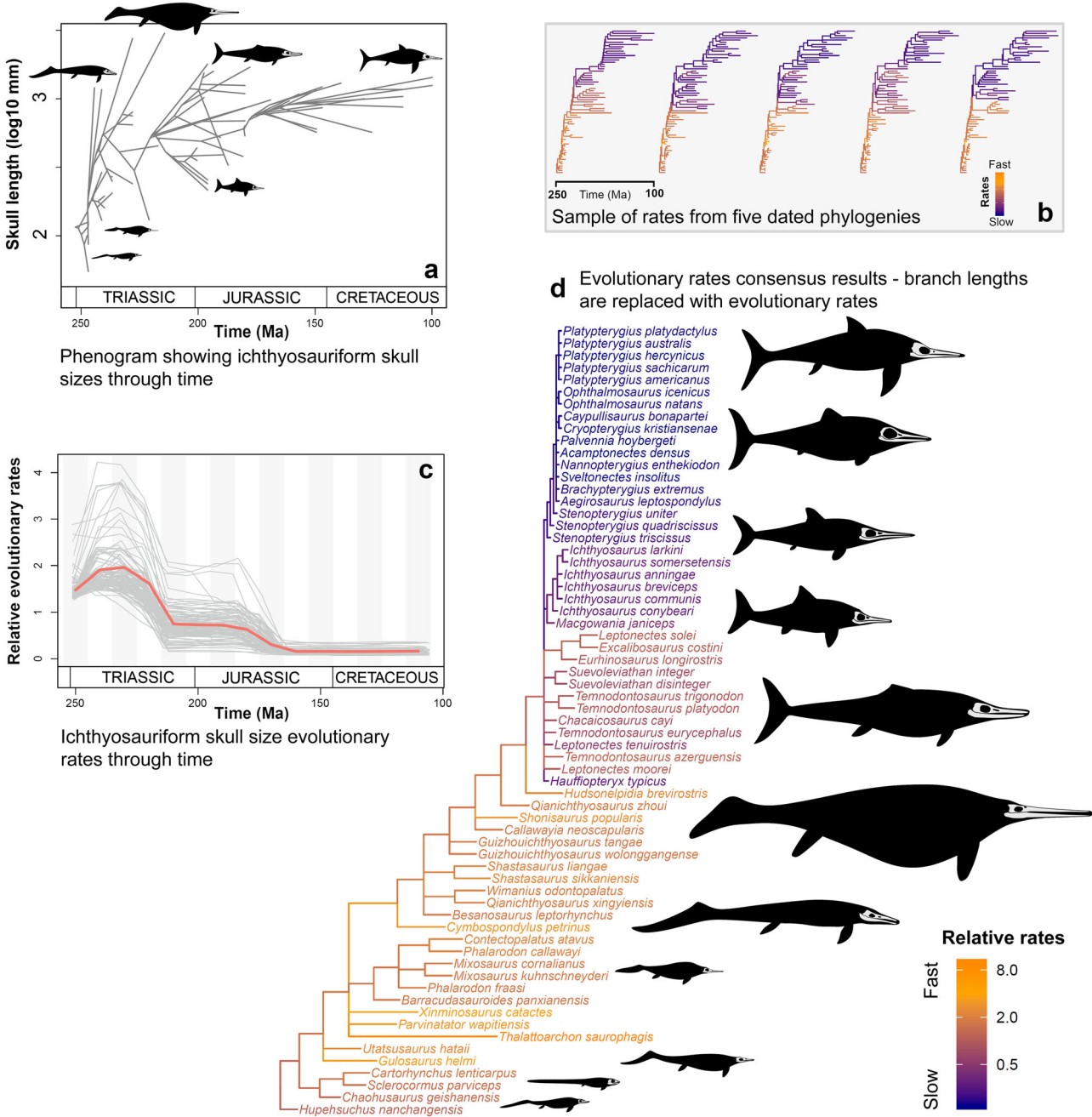

**Fig. 4 Rates of skull size evolution in Ichthyosauriformes. a** Phenogram showing the disparity of skull sizes through the Mesozoic. **b** Evolutionary rates results from five Hedman-dated phylogenies (out of 100 in total), with branches coloured by their rate value (warmer colours equal faster rates). **c** Evolutionary rates-through-time based on all 100 sampled Hedman-dated phylogenies, calculated in 16, 10 my time bins (red line is mean rate through time and each grey line represents an individual analysis). **d** Summary of evolutionary rates from all 100 Hedman-dated phylogenies with branches coloured with their rate value and branch lengths replaced with evolutionary rates (longer branches equal fast rates, shorter branches equal slow rates).

also removed. Morphospace occupation was visualised using only the first three principal coordinate axes from each ordinated distance matrix (Supplementary Fig. 6), but all axes were included in statistical tests. Additional measures of disparity—per-bin sum of variances, sum of ranges, centroid distance—were calculated using the R package dispRity[48] (Supplementary Fig. 4) with rarefaction. Sum of variances proves invariant to changes in bin populations: the mean value remained constant, although the error reduced, when increasing the number of taxa included (Supplementary Fig. 3). We tested for difference in morphospace occupation between time bins using pairwise PERMANOVA tests using the function adonis in the R package vegan[51]. To test for differences in disparity between bins, we used pairwise *t*-tests for each distance and disparity metric. In both cases, *p*-values were subsequently adjusted using the false discovery rate method[52]. Output CSV files from all statistical tests are included in Supplementary Code 1 DOI 10.5281/zenodo.3584386[58].

**Phylogenetic trees and time-scaling**. Time-scaled phylogenies with estimated branch lengths were required for evolutionary rates analyses. To account for uncertainty in ichthyosauriform phylogenetic relationships we used 100 phylogenetic trees, each with a different topology, from the Bayesian phylogenetic analyses of Moon[20]. For each of the 100 phylogenetic trees, a point age for each taxon was drawn from a uniform distribution between their FAD and LAD (Supplementary Table 3). Therefore, the 100 trees incorporated both phylogenetic and dating uncertainty. We implemented the Hedman time-scaling method[27], which uses Bayesian statistics to date nodes, and requires constraints derived from successive outgroup taxa ages[27,28]. The occurrence dates of the following outgroup taxa were used: *Claudiosaurus*, *Thadeosaurus*, *Hovasaurus*, *Captorhinus*, *Mesosaurus*, and *Petrolacosaurus* (Supplementary Table 2). We use a whole-tree extension of the Hedman algorithm using the R functions from Lloyd et al.[28] Note that the Hedman dating analyses do not always complete successfully, so the code in Supplementary Code 1 initially uses 120 trees

then downsamples to 100 successfully time-scaled trees. The minimum branch length method of time scaling was also used to compare the results in discrete rates analyses (Supplementary Methods, Supplementary Fig. 7).

**Discrete character evolutionary rates**. We analysed rates of discrete skeletal character evolution in a maximum-likelihood framework using the function DiscreteCharacterRate from Claddis[22]. For each of 100 time-scaled phylogenetic trees, we first used the rerootingMethod function[53] to estimate ancestral states across the trees from the character matrix of Moon[20] and then identify branches that have significantly higher or lower rates than collected across the tree using likelihood ratio tests[26]. Scripts from Close et al.[26] were modified for these analyses and to produce spaghetti plots that show the per-bin (both epoch- and equal-length) character change rates for each tree, and the mean character change rate across all trees.

**Skull size data and evolutionary rates**. Ichthyosauriformes had great size disparity and taxa ranged from <1 m (*Cartorhynchus, Mixosaurus*) to over 21 m (*Shastasaurus*) in total body length[32,34]. There is no agreed proxy for body mass in ichthyosauriforms. Body length could be a suitable proxy, however, completely preserved ichthyosauriforms (including complete caudal series) are relatively rare. Here we use maximum length of the skull as a general estimate for size (Supplementary Table 4). It is important to consider that ichthyosauriforms do show variation in skull proportions and the relative size of the skull compared to whole body length[32]. Nevertheless, we consider skull size to be an important component of morphological variation. In addition, it is more readily available to measure than total body length, and we are interested in broad-scale trends incorporating large magnitudes of size disparity. Maximum skull length was recorded for 64 ichthyosauriform species and the outgroup *H. nanchangensis*. Measurements were log$_{10}$ transformed prior to rates analyses.

Skull size evolutionary rates were analysed in a Bayesian framework using the variable-rates model in BayesTraits version 2.0.2[29] and R[23]. For 100 time-scaled phylogenetic trees, evolutionary rate heterogeneity was examined using a reversible jump Markov Chain Monte Carlo algorithm (rjMCMC) with default prior distributions. Each tree was run for 220 million iterations with parameters sampled every 10,000 iterations. The first 20 million iterations were discarded as burn-in. In brief, the variable-rates model detects rate shifts by rescaling branch lengths where phenotypic change deviates from that expected of a homogeneous Brownian motion model (variance proportional to branch lengths). The magnitude of stretching or compression (the rate scalar) indicates the magnitude of deviation from the background rate on the branch of interest[19]. Model fit was tested using Bayes factors (BFs), with marginal likelihoods for Brownian (homogeneous) and variable-rates (heterogeneous) models calculated using stepping-stone sampling[54], with 100 stones each run for 1000 iterations. Convergence was assessed based on the smallest effective sample size (ESS) and using the R package CODA[55]. Across the 100 phylogenies, the minimum ESS value was 3505.

The Variable Rates Post Processor[19] was used to extract the skull size evolutionary rate estimates. Here branch-specific rate values are based on the mean scalar parameter. We examined evolutionary rates for 100 time-scaled trees by colouring the branches with their rate value (the first five are shown in Fig. 4b, all trees are presented in the Supplementary Fig. 8). Results were summarised using a consensus tree from all 100 topologies, where branch lengths are replaced with evolutionary rates (Fig. 4d). To generate this consensus tree the branch lengths for all trees were substituted with the mean rate scalars, then the mean branch lengths from all 100 trees are calculated, ignoring edges that were not present in all trees (using R packages phytools[53] and ggtree[56]). Temporal evolutionary rate trends were calculated using the Variable Rates Post Processor[19] with 16, ~10-my time slices per tree. The tool also accounts for shared ancestry as implied by phylogeny[29,57]. Rates through time are plotted for each time bin based on all 100 trees and the mean rate.

**Reporting summary**. Further information on research design is available in the Nature Research Reporting Summary linked to this article.

## Data availability

Supplementary Figures, Supplementary Tables, and Supplementary Code can be found in the GitHub repository (https://github.com/benjaminmoon/ichthyosaur-macroevolution) and can be found at https://doi.org/10.5281/zenodo.3584386 [58]. Cladistic data is taken from Moon (2018) and is available with that paper. Phenomic data is included in the supplementary information and hosted in Zenodo and GitHub https://doi.org/10.5281/zenodo.3584386[58].

## Code availability

The authors declare that the code supporting the findings of this study is available at https://doi.org/10.5281/zenodo.3584386 [58].

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

## Acknowledgements

The authors are very grateful to Armin Elsler (University of Bristol) for sharing R code that greatly improved the efficiency of processing and plotting outputs from BayesTraits, and to Susana Gutarra Diaz for providing artwork used in Fig. 4. Part of this work was carried out using the computational facilities of the Advanced Computing Research Centre, University of Bristol (http://www.bris.ac.uk/acrc/). B.C.M. was funded by Leverhulme Trust Research Project Grant RPG-2015-126 and BETR grant NE/P013724/1; T.L.S. was funded by BETR grant NE/P013724/1 and ERC grant 788203 (INNOVATION).

## Author contributions

B.C.M. led the project and conceived the study. B.C.M. and T.L.S. ran the analyses, wrote, and approved the manuscript.

## Competing interests

The authors declare no competing interests.
