## [Peer Review File · Communications Biology]

REVIEWERS' COMMENTS:

Reviewer #1 (Remarks to the Author):

This is an excellent, well-written, thoroughly executed and smartly illustrated study that rigorously analyses patterns of disparity and diversity throughout the evolution of ichthyosaurs. It will be of great interest to those working on marine reptiles, as well as all those interested in clade dynamics. I have nothing of any significance to criticise, and recommend publication more or less as the manuscript stands - in many ways it is a model.

Matt Wills

Reviewer #2 (Remarks to the Author):

Dear editor,

Please find my comments in the attached pdf.

Best regards,
Thomas Guillerme

Reviewer #3 (Remarks to the Author):

This is a well written, well executed and very scholarly contribution to macroevolutionary studies of ichthyosaurs, one of the most iconic and best studied groups of Mesozoic amniote, and an important taxon for our understanding of models of morphological and clade diversification during key episodes of ecological innovation. The authors have rapidly made a name for themselves in the landscape of analytical palaeontology and have consolidated expertise in the areas covered by the research. Their paper is a detailed tour-de-force through the analysis of ichthyosaur morphological diversification in a three-pronged approach to the study of evolutionary rates, disparity, and morphological space occupation. It is well written and provides ample documentation for data and result reproducibility. I have no qualms with this paper. The only truly minor remark is about the phrasing of the title. I am not sure why the authors use the expression "fish-shaped reptiles". I would imagine "ichthyosaurs" is perfectly valid and appealing, but I do not wish to question their choice. I am very happy to recommend this paper for publication pretty much in its current format.

Reviewer #2

Review COMMSBIO 19-1428

2019-11-11

Dear Editor,

Please find below my comments on the manuscript entitled “Early high rates and disparity in the evolution of fish-shaped reptiles”.

In this paper, the authors look at the evolution of the diversity of morphologies (disparity) in ichthyosaur looking at both disparity through time and morphological diversity rates. They found that ichthyosaurs display a classic “Early burst” mode of morphological diversity evolution with different shapes accumulating faster in the clade’s early history.

This is a great paper with a solid and well used methodology. Although the paper does not bring some major new dataset (the data is based on Moon 2018 [ref 20]) nor major methodological advances (the analysis is based on Close et al 2015 [ref 46]) it has a really interesting and solid narrative and will certainly be of great interest to a general audience in biologists (either for the natural history insights or for emulating disparity-through-time analysis).

The authors did a great job with sharing the code and the data to reproduce this paper. Although this should be a standard in science, it is unfortunately done well and in depth way to rarely. Well done!

I have no major concern about the data and the analysis but I have a couple of major comments and several minor suggestions that could improve the already high quality of this manuscript.

Major Comments

Clarity

Although I do understand that the introduction-results-discussion-methods format is probably a requirement of the journal - which is OK in some circumstances - I think here it is a disservice to the paper (from my point of view and because of the solid methodology of the authors). I only grasped the extant of the authors’ work and significance at my second reading, after I already read the methods section. If the format is a requirement of the journal, I encourage the authors to rewrite the results section and explain more of the methodology in there, otherwise, I suggest the authors to revert back to the more classic introduction-methods-results-discussion format.

Number of bootstraps

The authors use 10000 bootstrap pseudo-replications to estimate their disparity metric’s variance. I think this number is pretty high and could seriously inflate the false positive error (type I), especially when using a parametric t-test to test the difference in disparity between bins. I suggest the authors rerun the analysis using a lower number of bootstraps (say 1000 or 500).

Disparity through time

The authors demonstrate an interesting effect of the time subsampling methods with clear different results when using geological time binning or equal size time binning. Out of personal interest, I wonder what time-slicing methods will tell under various explicit evolutionary models (Guillerme and Cooper 2018).

Minor suggestions

- 1.8-9: I don't see the connection between clades' early history and mass extinctions. Although maybe the authors meant that it is important to understand how clades evolve after a mass extinction?
- 1.15-16: "complete phylogenetic tree"; maybe change to "most complete to date"?
- 1.45: I think it could be important to determine what the authors mean by niche space. Although many paleobiologists have the same general idea, it definitely differs between sub-disciplines in biology.
- 1.85: I suggest changing "a larger morphospace" to "a larger part of the morphospace".
- 1.141: I suggest spelling out Early Burst here to strengthen the impact of the authors' claim.
- 1.144: This "ecological niches" claim always makes me uncomfortable in disparity papers. Although the whole palaeobiology and macroevolution community (myself included) has this intuition that disparity is probably linked to ecological niches, it has yet to be tested (and is heavily debated in ecology). I suggest being more cautionary and using a conditional statement or dropping the niche story altogether.
- 1.168: This is in line with Chen & Benton 2012 post P-T extinction recovery time estimates.
- 1.176-184: It would be interesting to re-run the disparity analysis with only the characters from reference 29 (or similar) to confirm/infirm whether using cladistic characters give a more or less gradual image of ichthyosaurs' morphological evolution.
- 1.233: What do the authors mean by "total knowledge"?
- 1.258-259: For improving clarity, I suggest the authors also write down the disparity metric mathematical definition (like in ref 46).
- 1.264: how many taxa with incomparable characters were removed? And what algorithm was used to remove them?
- 1.268: Why "primarily"? The supplementary materials seems to contain *only* representation of the three first axis (which is not problematic *per se*).
- 1.272: Did the authors also run a rarefaction analysis for the pairwise dissimilarity metric?
- 1.274-275: "morphospace separation" do the authors mean difference in morphospace occupancy?
- 1.281-282: I suggest using "branch length" rather than "branch duration".
- 1.282: why using 120 trees? This number seems odd (although not especially wrong).
- 1.282: it is unclear to me whether 108 Bayesian analysis converged or whether 108 trees (out of the 120) are part of the same analysis and presumably the trees past the burnin phase (or I'm missing the "converged" point).
- 1.287: So these 120 trees don't have the same topology as well?
- 1.297: Again, I'm confused whether 120 or 108 trees are used at all time. Or, if not, when and why do the authors decide to use 120 or 108 trees.
- 1.298-299: What method was used to estimate the ancestral states? Also, this is a semantic point but the authors might prefer using *estimates* rather than *reconstructions*. "Reconstructions" conveys more an idea of solid ancestral states whereas "estimates" (with error) are more closer to reality.
- 1.310-315: This is a cautious and solid justification!
- 1.319: I'm still confused about the number of trees used (why a 100 now?) I suggests the authors use these 100 trees across the whole analysis.
- Figure 1: I suggest relabeling the y-axis label as Pairwise Euclidean distance (rather than dissimilarity - if that's indeed the used metric).

- Figure 3: Although the green/pink colour scheme works in figure 1, I had hard time to distinguish the two colours in this figure. I suggest the authors change it to something more contrasted (and still colour blind friendly - I suggest orange and blue).
- Figure 4: This is a great figure! However, it's missing a scale bar for branch lengths (since the branch lengths are rate, I assume a timescale will do).
- Code: 0-Moon_Stubbs-run_analyses.R: 1.27: I suggest adding an automatic package version check for Claddis:

```
if (!require(Claddis)) devtools::install_github("graemetlloyd/Claddis", ref = "4f7f1bf")
if(packageVersion("Claddis") != "0.2") {
  ## Downgrading the Claddis package
  remove.packages("Claddis")
  devtools::install_github("graemetlloyd/Claddis", ref = "4f7f1bf"); library(Claddis)
}
```

- Code: disparity_functions.R: I've noticed that the authors mean to measure the 95% CI interval (e.g. line 44-46) but the code actually measures the 90% CI interval ($0.95-0.05 = 0.9$). I suggest changing the comment or the code (e.g. using 0.025 and 0.975).

Best regards,

Thomas Guillaume

** References **

Guillaume T, Cooper N. Time for a rethink: time sub-sampling methods in disparity-through-time analyses. *Palaeontology*. 2018 Jul;61(4):481-93.

Chen, Z.-Q. and M. J. Benton. 2012. The timing and pattern of biotic recovery following the end-Permian mass extinction. *Nature Geoscience* 5:375–383.

Response to Reviewers

Reviewer #1 (Remarks to the Author):

This is an excellent, well-written, thoroughly executed and smartly illustrated study that rigorously analyses patterns of disparity and diversity throughout the evolution of ichthyosaurs. It will be of great interest to those working on marine reptiles, as well as all those interested in clade dynamics. I have nothing of any significance to criticise, and recommend publication more or less as the manuscript stands - in many ways it is a model.
Matt Wills

We thank the reviewer for giving their time to review our typescript, and for their kind comments.

Reviewer #2

Dear Editor,

Please find below my comments on the manuscript entitled “Early high rates and disparity in the evolution of fish-shaped reptiles”.

In this paper, the authors look at the evolution of the diversity of morphologies (disparity) in ichthyosaur looking at both disparity through time and morphological diversity rates. They found that ichthyosaurs display a classic “Early burst” mode of morphological diversity evolution with different shapes accumulating faster in the clade’s early history.

This is a great paper with a solid and well used methodology. Although the paper does not bring some major new dataset (the data is based on Moon 2018 [ref 20]) nor major methodological advances (the analysis is based on Close et al 2015 [ref 46]) it has a really interesting and solid narrative and will certainly be of great interest to a general audience in biologists (either for the natural history insights or for emulating disparity-through-time analysis).

The authors did a great job with sharing the code and the data to reproduce this paper. Although this should be a standard in science, it is unfortunately done well and in depth way to rarely. Well done!

I have no major concern about the data and the analysis but I have a couple of major comments and several minor suggestions that could improve the already high quality of this manuscript.

The reviewer has admirably summed up our aims and methods and we thank them for their time and kind response to our work.

Major Comments

Clarity

Although I do understand that the introduction-results-discussion-methods format is probably a requirement of the journal - which is OK in some circumstances - I think here it is a disservice to the paper (from my point of view and because of the solid methodology of the authors). I only grasped the extent of the authors’ work and significance at my second reading, after I already read the methods section. If the format is a requirement of the

journal, I encourage the authors to rewrite the results section and explain more of the methodology in there, otherwise, I suggest the authors to revert back to the more classic introduction-methods-results-discussion format.

The order of sections is a requirement of the journal and is specified in the instructions from the copy editor. We have taken on board the reviewer's suggestions and added some further description of the methods used. These are indicated in the revised typescript and at the beginning of each of the results subsections.

Number of bootstraps

The authors use 10000 bootstrap pseudo-replications to estimate their disparity metric's variance. I think this number is pretty high and could seriously inflate the false positive error (type I), especially when using a parametric t-test to test the difference in disparity between bins. I suggest the authors rerun the analysis using a lower number of bootstraps (say 1000 or 500).

We accept that the high number of bootstrap replicates may result in increased error. We have changed this number to 500 throughout and thank the reviewer for bringing this to our attention.

Disparity through time

The authors demonstrate an interesting effect of the time subsampling methods with clear different results when using geological time binning or equal size time binning. Out of personal interest, I wonder what time-slicing methods will tell under various explicit evolutionary models (Guillerme and Cooper 2018).

While our results have differences based on the binning scheme, we feel that extending our study to include time-slicing under numerous possible models would be a substantial piece of work and extend beyond the scope of this contribution. We do, however, consider it worthwhile exploring in future studies.

Minor suggestions

- l.8-9: I don't see the connection between clades' early history and mass extinctions. Although maybe the authors meant that it is important to understand how clades evolve after a mass extinction?
Yes this is how we mean and we've changed this sentence to make this clearer.
- l.15-16: "complete phylogenetic tree"; maybe change to "most complete to date"?
Changed to 'most complete phylogenetic tree to date'.
- l.45: I think it could be important to determine what the authors mean by niche space. Although many paleobiologist have the same general idea, it definitely differs between sub-disciplines in biology.
We have added the clarifier 'the specific size, habitat, feeding, locomotory and other traits of an organism'.

- l.85: I suggest changing “a larger morphospace” to “a larger part of the morphospace”.
Change made.
- l.141: I suggest spelling out Early Burst here to strengthen the impact of the authors’ claim.
We have done this.
- l.144: This “ecological niches” claim always makes me uncomfortable in disparity papers. Although the whole palaeobiology and macroevolution community (myself included) has this intuition that disparity is probably linked to ecological niches, it has yet to be tested (and is heavily debated in ecology). I suggest being more cautionary and using a conditional statement or dropping the niche story altogether.
We thank the reviewer for their suggestion. In this case we use this in concert with other research on the ecological diversification of ichthyosaurs (ref. 22) as so treat this as a comparison with the adaptation into niches rather than necessarily drawing a direct comparison. A reference to this has been added here for clarity.
- l.168: This is in line with Chen & Benton 2012 post P-T extinction recovery time estimates.
We have added this reference as new number 29.
- l.176-184: It would be interesting to re-run the disparity analysis with only the characters from reference 29 (or similar) to confirm/infirm whether using cladistic characters give a more or less gradual image of ichthyosaurs’ morphological evolution.
We thank the reviewer for the suggestion, but think that the data set in ref 29 [revised numbering 22] does not present a reasonable use considering the relatively few characters and states due to ‘lumping’ into broad ecological groupings.
- l.233: What do the authors mean by “total knowledge”?
We have changed this to ‘all reasonably assigned material’.
- l.258-259: For improving clarity, I suggest the authors also write down the disparity metric mathematical definition (like in ref 46).
We have added this.
- l.264: how many taxa with incomparable characters were removed? And what algorithm was used to remove them?
114 ichthyosauriform taxa and the outgroup were removed leaving 100 taxa. TrimMorphDistMatrix in the package Claddis was used to remove them. We have added this to the typescript.
- l.268: Why “primarily”? The supplementary materials seems to contain *only* representation of the three first axis (which is not problematic *per se*).
We have removed ‘primarily’.

- I.272: Did the authors also ran a rarefaction analysis for the pairwise dissimilarity metric?
We have added this.
- I.274-275: “morphospace separation” do the authors mean difference in morphospace occupancy?
We’ve changed this to ‘difference in morphospace occupation’.
- I.281-282: I suggest using “branch length” rather than “branch duration”.
We have made this change.
- I.282: why using 120 trees? This number seems odd (although not especially wrong).
- I.282: it is unclear to me whether 108 Bayesian analysis converged or whether 108 trees (out of the 120) are part of the same analysis and presumably the trees past the burnin phase (or I’m missing the “converged” point).
- I.287: So these 120 trees don’t have the same topology as well?
- I.297: Again, I’m confused whether 120 or 108 trees are used at all time. Or, if not, when and why do the authors decide to use 120 or 108 trees.
- I.319: I’m still confused about the number of trees used (why a 100 now?) I suggests the authors use these 100 trees across the whole analysis.
In the case of using the Hedman method to time-scale trees, the functions used in R infrequently result in errors produced doing the analysis in R; we are uncertain exactly why this happens. We initially input 120 trees into the time-scaling analysis to ensure at least 100 trees result. In the revised version we have used 100 time-scaled trees for all subsequent analyses and changed the typescript to reflect that.
- I.298-299: What method was used to estimate the ancestral states? Also, this is a semantic point but the authors might prefer using *estimates* rather than *reconstructions*. “Reconstructions” conveys more an idea of solid ancestral states whereas “estimates” (with error) are more closer to reality.
We modified the sentence to say ‘estimate’ ancestral states rather than ‘reconstruct’. Ancestral states were estimated using the rerootingMethod function that is built into the DiscreteCharacterRate Claddis function.
- I.310-315: This is a cautious and solid justification!
Thank you for comment. We think this is reasonable given there is some variation in skull length that reflects habits and ecology.
- Figure 1: I suggest relabeling the y-axis label as Pairwise Euclidean distance (rather than dissimilarity - if that’s indeed the used metric).
We have changed this to ‘Pairwise maximum observed rescaled distance’ as this is the distance metric used.
- Figure 3: Although the green/pink colour scheme works in figure 1, I had hard time to distinguish the two colours in this figure. I suggest the authors change it to something more contrasted (and still colour blind friendly - I suggest orange and blue).

We have changed this to an orange/yellow-blue combination using the colorspace package in R, also attempting to keep a brightness difference for reproduction in greyscale.

- Figure 4: This is a great figure! However, it's missing a scale bar for branch lengths (since the branch lengths are rate, I assume a timescale will do).
Yes, this is correct for part b and we have added a timescale for this part of the plot. In part d the branch lengths are scaled by the rate so their lengths correspond to rate of evolution (not geological time) and this is represented in the key. The branch lengths in d are all relative so do not require a specific time scale.
- Code: 0-Moon_Stubbs-run_analyses.R: l.27: I suggest adding an automatic package version check for Claddis:
Thank you for the code suggestion, this has been added.
- Code: disparity_functions.R: I've noticed that the authors mean to measure the 95% CI interval (e.g. line 44-46) but the code actually measures the 90% CI interval ($0.95 - 0.05 = 0.9$). I suggest changing the comment or the code (e.g. using 0.025 and 0.975).
Thank you for pointing this out, the code has been changed.

Best regards,
Thomas Guillerme

Reviewer #3 (Remarks to the Author):

This is a well written, well executed and very scholarly contribution to macroevolutionary studies of ichthyosaurs, one of the most iconic and best studied groups of Mesozoic amniote, and an important taxon for our understanding of models of morphological and clade diversification during key episodes of ecological innovation. The authors have rapidly made a name for themselves in the landscape of analytical palaeontology and have consolidated expertise in the areas covered by the research. Their paper is a detailed tour-de-force through the analysis of ichthyosaur morphological diversification in a three-pronged approach to the study of evolutionary rates, disparity, and morphological space occupation. It is well written and provides ample documentation for data and result reproducibility. I have no qualms with this paper.

We thank the reviewer too for commenting on our manuscript and for kindly giving us such favourable reviews.

The only truly minor remark is about the phrasing of the title. I am not sure why the authors use the expression "fish-shaped reptiles". I would imagine "ichthyosaurs" is perfectly valid and appealing, but I do not wish to question their choice. I am very happy to recommend this paper for publication pretty much in its current format.

The reviewer's comment is valid and we used 'fish-shaped reptiles' based on other, similarly-aimed recent works, but are happy to take the recommendation of the author and copy editor in modifying our title to:
'Early high rates and disparity in the evolution of ichthyosaurs'.